# Development of an Eco-Friendly Nanogel Incorporating *Pectis brevipedunculata* Essential Oil as a Larvicidal Agent Against *Aedes aegypti*

**DOI:** 10.3390/pharmaceutics16101337

**Published:** 2024-10-18

**Authors:** Estela Mesquita Marques, Raiene Lisboa Rocha, Clenilma Marques Brandão, Júlia Karla Albuquerque Melo Xavier, Marcos Bispo Pinheiro Camara, Caritas de Jesus Silva Mendonça, Roberto Batista de Lima, Melissa Pires Souza, Emmanoel Vilaça Costa, Renato Sonchini Gonçalves

**Affiliations:** 1Laboratory of Chemistry of Natural Products, Department of Chemistry, Federal University of Maranhão (UFMA), São Luís 65080-805, Brazil; estela.marques@discente.ufma.br (E.M.M.); raiene.rocha@discente.ufma.br (R.L.R.); julia.karla@ufma.br (J.K.A.M.X.); quimarcosbispo@hotmail.com (M.B.P.C.); 2Department of Chemistry, Federal Institute of Maranhão (IFMA), São Luis 65075-441, Brazil; clenilma.brandao@ifma.edu.br; 3Center for Fuels, Catalysis, and Environment (NCCA), Department of Chemistry, Federal University of Maranhão (UFMA), São Luís 65080-805, Brazil; cjs.mendonca@ufma.br; 4Department of Chemistry, Federal University of Maranhão, São Luís 65080-805, Brazil; rb.lima@ufma.br; 5Postgraduate Program in Chemistry, Federal University of Amazonas (UFAM), Manaus 69080-900, Brazil; mel.souza.1215@gmail.com (M.P.S.); emmanoelvc@gmail.com (E.V.C.); 6Department of Chemistry, Federal University of Amazonas (UFAM), Manaus 69080-900, Brazil

**Keywords:** thermoresponsive nanogel, *Pectis brevipedunculata* essential oil, *Aedes aegypti*

## Abstract

Background/Objectives: Arboviruses, transmitted by mosquitoes like *Aedes aegypti*, pose significant public health challenges globally, particularly in tropical regions. The rapid spread and adaptation of viruses such as Dengue, Zika, and Chikungunya have emphasized the need for innovative control methods. Essential oils from plants, such as *Pectis brevipedunculata* (Gardner) Sch.Bip. (*Pb*), have emerged as potential alternatives to conventional insecticides. Methods: In this work, we developed an eco-friendly nanogel using a low-energy, solvent-free method, incorporating the copolymer F127 and Carbopol 974p, enriched with a high concentration of essential oil from *Pb* (EO*Pb*). The resulting nanogel displayed excellent physical stability, maintained under varying temperature conditions. Characterization techniques, including FTIR and DLS, confirmed the stable incorporation of EO*Pb* within the nanogel matrix. Results: The in vitro assays against *Aedes aegypti* larvae revealed that at 500 μg/mL, the mortality rates were 96.0% ± 7.0 after 24 h and 100.0% ± 0.0 after 48 h. The positive control group treated with temefos, achieved 100% mortality at both time points, validating the experimental conditions and providing a benchmark for assessing the efficacy of the nGF2002*Pb* nanogel. Conclusions: These results indicate that nGF2002*Pb* demonstrates a pronounced concentration-dependent larvicidal effect against *Aedes aegypti*, offering an innovative and sustainable approach to arbovirus vector control.

## 1. Introduction

Arboviruses, transmitted primarily by arthropods such as mosquitoes and ticks, represent a global public health concern, particularly in tropical and subtropical regions [1,2]. These viruses, including notable members like Dengue, Zika, and Chikungunya, have shown an alarming capacity for rapid spread and adaptation to new environments and hosts [3,4]. The resurgence of arbovirus outbreaks, coupled with the emergence of novel strains [5,6], poses a substantial challenge for public health systems [7], necessitating the development of effective therapeutic strategies and vaccines [8].

Recent advancements in pharmaceutical research have focused on identifying antiviral compounds, developing vaccine candidates [9], and exploring innovative delivery systems to enhance efficacy and stability [10,11]. Given the complexity of arbovirus transmission dynamics and the potential for co-infections [12,13], integrated approaches that combine vector control with pharmacological interventions are essential for mitigating the impact of these viruses on global health [14,15]. Among the most significant vectors is *Aedes aegypti*, the mosquito responsible for spreading these arboviruses across tropical and urbanized regions [16,17]. The species’ adaptability to human environments, daytime biting habits, and resistance to traditional insecticides complicate control efforts [18,19]. While conventional chemical insecticides have been widely used, their effectiveness is declining due to the growing resistance in mosquito populations [20,21]. This challenge has prompted a shift toward more sustainable and eco-friendly approaches, with an increasing focus on natural products as alternatives [22,23,24].

Plant-based compounds, essential oils (EOs), and botanical extracts are being extensively researched for their potential to act as larvicides, repellents, and adulticides against *Aedes aegypti* [25,26]. For example, EO from plants like *Cymbopogon citratus* (D.C.) Stapf (lemongrass), *Azadirachta indica* A. Juss. (neem), and *Eucalyptus globulus* Labill. have shown promising results in disrupting the mosquito’s life cycle and reducing its ability to transmit diseases [27,28,29]. These natural products are environmentally benign and less likely to contribute to resistance, making them an appealing strategy in integrated vector management [30,31,32]. The continued exploration of these natural compounds, coupled with advancements in formulation and delivery methods, offers a promising avenue for sustainable mosquito control that could significantly reduce the incidence of arbovirus-related diseases [33,34]. Integrating nanoformulations in mosquito control strategies presents a transformative approach to combating *Aedes aegypti* [35,36]. Nanoformulations offer enhanced stability, targeted delivery, and controlled release of active agents, thereby increasing efficacy while minimizing environmental impact [37].

By encapsulating natural products within nanocarriers, these formulations can achieve prolonged activity and reduce the frequency of application. Moreover, the precision targeting of *Aedes aegypti* populations with these nano-based systems has the potential to disrupt the transmission cycle of arboviruses, leading to a substantial reduction in disease prevalence [38]. This innovative approach underscores the synergy between nanotechnology and vector control, offering a promising avenue for the development of next-generation solutions in the fight against vector-borne diseases [39].

Nanogels and nanoemulsions have emerged as a versatile platform for delivering bioactive compounds in vector control [40]. Their three-dimensional polymeric network allows for high loading capacity and controlled release of active ingredients, making them particularly suitable for sustained mosquito control interventions [41]. When formulated with essential oils or other natural products, nanogels can provide a protective matrix that enhances the stability and bioavailability of these agents, while reducing their volatility and degradation. This ensures prolonged efficacy against *Aedes aegypti* and can help in overcoming challenges related to the rapid degradation of traditional insecticides [42]. Moreover, the biocompatibility and tunable properties of nanogels and nanoemulsions allow for the design of targeted delivery systems that can minimize non-target effects and environmental impact [43].

Aiming to broaden the existing research on the use of nanoformulations containing natural products as an efficient alternative for combating arbovirus vectors, we present, for the first time, the development of the thermoresponsive nanogel nG2002*Pb* using a low-energy and solvent-free method. This nanogel predominantly contains the copolymer F127, a material already approved by the FDA for various pharmacological uses, along with a smaller amount of the polymer 974p and a high EO*Pb* concentration. *Pectis brevipedunculata*, belonging to the Asteraceae family, is a herbaceous plant commonly found in tropical and subtropical regions of the Americas, particularly in countries such as Brazil, Argentina, and Uruguay [44]. Known for its potential medicinal properties, it has been traditionally used in various cultures for its therapeutic benefits, including anti-inflammatory and antimicrobial effects. This species is characterized by its aromatic foliage and distinctive yellow flowers, making it a significant subject of study in ethnopharmacology and natural product research. This focus is particularly relevant in the development of natural medicines and in understanding its ecological role. The *Pectis* genus, known for its high content in monoterpenes, holds significant potential as a source of natural larvicides [45]. Monoterpenes such as neral, geranial, α-pinene, and limonene, present in OE*Pb*, exhibit strong larvicidal activity against mosquito species like *Aedes aegypti* [46,47]. These compounds disrupt essential physiological processes in larvae, leading to high mortality rates [48]. Leveraging the monoterpene-rich *Pectis* genus as a natural larvicide presents an eco-friendly alternative to synthetic chemicals, enhancing integrated vector management strategies to control disease-transmitting mosquitoes.

In this work, accelerated stability and shelf-life tests were performed to assess the physical and chemical stability of nG2002*Pb*. Characterization through FTIR and DLS techniques confirmed the successful incorporation of EO*Pb* within the nanogel matrix. The in vitro assays conducted on *Aedes aegypti* larvae demonstrated a markedly high mortality rate after 48 h of exposure, underscoring the effectiveness of the treatment. This study highlights nGF2002*Pb* as a novel, efficient, and eco-friendly alternative, showcasing its substantial larvicidal activity against *Aedes aegypti* and positioning it as a promising and sustainable solution for arbovirus vector control.

## 2. Materials and Methods

### 2.1. Materials

Pluronic F127 (poly(ethylene oxide)–poly(propylene oxide)–poly(ethylene oxide)) triblock copolymer (MW = 12,600 g/mol; (EO_99_(PO)_67_(EO)_99_)), ultrapure water, anhydrous sodium sulfates, deuterated chloroform (CDCl_3_), and alkane standard mixture were commercially acquired from the Merck company (Rahway, NJ, USA). Carbopol^®^ 974p NF polymer was kindly provided by IMCD Brazil (São Paulo, SP, Brazil). *Aedes aegypti* eggs were supplied by the Moscamed biofactory (São Francisco, Juazeiro, BA, Brazil) and temefos was provided by sanitary surveillance agents (São Luís, MA, Brazil). The reagents were used as received, without further purification.

### 2.2. Plant Material

The herbaceous plant *P. brevipedunculata* was collected from the Universidade Federal do Maranhão (UFMA) campus in São Luís, MA, Brazil, at coordinates 2°33′20.5″ S and 44°18′32.7″ W. A voucher specimen (no. 5287) was deposited in the Herbarium Rosa Mochel (SLUI) at Universidade Estadual do Maranhão (UEMA), São Luís, MA, Brazil. The plant collection adhered to Brazilian biodiversity protection laws (SisGen registration AAFB38B).

### 2.3. Extraction Procedure

Hydrodistillation using a Clevenger-type apparatus was employed to extract essential oil from *P. brevipedunculata* (EO*Pb*). Plant material (300 g) was air-dried and cut into small pieces using pruning shears to facilitate the hydrodistillation extraction. Distilled water (500 mL) was added to the flask, and the hydrodistillation process was maintained for 2.5 h after the reflux began. Following the extraction period, the oil/water (O/W) phase mixture was centrifugated at 3500 rpm for 10 min at a controlled temperature of 25 °C. Anhydrous sodium sulfate was used to eliminate any remaining traces of water, and the (EO*Pb*) was yielded at 0.81% of the initial plant material weight (Figure 1).

### 2.4. CG-MS Analysis

Analyses were conducted utilizing a CG-2010 SE Shimadzu (Kyoto, Japan) equipped with a flame ionization detector (FID) and electronic integrator. Compound separation was achieved using an RXi-1MS fused capillary column (30 m × 0.25 mm × 0.25 mm film thickness) coated with 5%-phenyl-arylene–95% dimethylpolysiloxane. Helium served as the carrier gas at a 1.0 mL/min flow rate. The column temperature program followed a sequence of 40 °C at 4 min, ramping up at a rate of 4 °C/min to 240 °C, then at 10 °C/min to 280 °C, with a hold at 280 °C for 2 min. Injector and detector temperatures were maintained at 250 and 280 °C, respectively. Samples (10 mg/mL in CH_2_Cl_2_) were injected with a 1:50 split ratio. Retention indices were established using a standard solution of n-alkanes (C_10_-C_40_), while peak areas and retention times were determined using an electronic integrator. The relative amounts of individual compounds were calculated from GC peak areas without FID response factor correction. GC–MS analyses were performed using a QP2010 SE CG-MS system Shimadzu (Kyoto, Japan) equipped with an AOC-20i auto-injector. MS spectra were acquired at 70 eV with scan intervals of 0.5 s and fragments ranging from 40 to 550 Da. Experimental conditions mirrored those of the GC analysis. EO components were identified by comparing GC peak retention times with standard compounds run under identical conditions, and by comparing retention indices and MS spectra with those reported in the literature and stored in the ADAMS and FFNSC libraries.

### 2.5. FTIR and UV-Vis Analysis

Fourier transform infrared spectroscopy (FTIR) analyses in reflectance mode were carried out using an FTIR Tracer-100 spectrophotometer Shimadzu (Kyoto, Japan). Samples of the materials nGF2002 and nGF2002*Pb* were lyophilized and pelletized with KBr, while a small amount of pure OE*Pb* material was analyzed in attenuated total reflectance (ATR) mode. For the ATR-FTIR analysis, the spectrometer was equipped with a horizontal ATR accessory, and a ZnSe crystal window (PIKE Technologies, Fitchburg, WI, USA) was used. Spectra were collected over the range of 400 to 4000 cm^−1^ with a resolution of 8 cm^−1^ and 50 scans. The sample spectra were obtained by first spreading the sample evenly on the ATR crystal surface, acquiring the spectrum, and then cleaning the crystal window with hexane and acetone before collecting subsequent spectra. Electronic absorption spectra of nanogels at a concentration of 5.0 × 10^−3^ g/mL in water were recorded at room temperature using a UV-Vis 1800 spectrophotometer (Shimadzu).

### 2.6. NMR Analysis

Nuclear magnetic resonance (NMR) spectra of the EO*Pb*, ^1^H and ^13^C NMR) were acquired on a BRUKER Avance III HD spectrometer (Billerica, MA, USA), operating at 11.75 Tesla (500.13 MHz for ^1^H NMR and 125.76 MHz for ^13^C NMR). The sample was dissolved in deuterated chloroform (CDCl_3_) and chemical shifts were expressed in parts per million (ppm) relative to tetramethylsilane (TMS), used as an internal reference standard with δ 0.00 (0.00 ppm).

### 2.7. Particle Size and ζ Potential

Average hydrodynamic diameters (D*_H_*) and ζ potential analyses were determined using dynamic light scattering (DLS) in water at 25 °C, with a 40 mW semiconductor laser at 658 nm, utilizing a Litesizer 500 Anton Paar GmbH instrument, Module BM 10 (Styria, Austria). The D*_H_* measurement was performed using a quartz cuvette of 3.0 mL. The ζ potential was performed using a low-volume cuvette (Univette). All the measurements were performed in triplicate (mean ± SD).

### 2.8. Preparation of Nanogel Formulations

The nanogel formulations were prepared following the experimental procedure described by Schmolka using the cold technique [49]. F127 copolymer was slowly added to cold distilled water and kept in an ice bath (5–10 °C), and the mixture was kept under gentle stirring to allow each flake of the copolymer to hydrate in the solution. After this period, Carbopol 974p was gradually added, and the solution was gently stirred at 5–10 °C until complete dissolution was achieved. EO*Pb* was then added to the solution drop by drop, keeping the solution under continuous stirring for 30 min. To ensure complete solubility of the ingredients, the solution was kept quiescent in the refrigerator at 5 °C overnight (Figure 2).

### 2.9. Stability Assay of Nanogel Formulations

To investigate the influence of temperature on the physical and chemical stability properties of nanogel formulations, accelerated stability tests were performed following the ANVISA Cosmetics Stability Guide and the US Pharmacopeia [50]. One milliliter of the formulation was centrifuged at 3000 rpm for 30 min at 25 ± 1 °C. The formulation was stored throughout the study both at room temperature (25 ± 3 °C) and under controlled refrigeration (5 ± 3 °C) (temperature was monitored daily). Therefore, the formulations were subjected to 7 cycles of 24 h at 5 °C and 24 h at 25 °C. Thus, physical stability was determined by observing parameters such as homogeneity, phase separation, and organoleptic characteristics (appearance, color, and odor). The chemical stability of the nanogels was evaluated by GC–MS analysis by comparing the concentration of EO*Pb* in the formulations immediately after nanogel preparation (day 0) and subsequently at intervals of 7 days, totaling 7 analysis points.

### 2.10. In Vitro Assays Against Aedes aegypti Larvae

The larvicidal profile of the nanogel nGF2002*Pb* was evaluated against *Aedes aegypti* larvae, following a protocol adapted from the WHO guidelines [51]. The hatching and larval-rearing procedures followed a methodology adapted by Amaral et al. and Carvalho et al. [52,53]. For the assays, 10 third-instar larvae were immersed in 20 mL of a nGF2002*Pb* formulation and in three control groups, all placed in polystyrene cups with a final volume of 50 mL. Five concentrations in EO*Pb* were applied: 5, 50, 100, 250, and 500 μg/mL, with 10 repetitions for each concentration, totaling 50 experiments and a sample size of 500 larvae. Additionally, 10 repetitions were conducted for the control group using mineral water for the hatching and larval growth stage and for the negative control with nGF2002, totaling 20 experiments and a sample size of 200 larvae. For the positive control, the commercial larvicide, temefos Fersol was used. Ten replicates were conducted, totaling 10 experiments and a sample size of 100 larvae. The assays were monitored over a 24–48 h period to determine the percentage of live and dead larvae, with the mortality rate calculated as a % of the mean and standard deviations.

### 2.11. Statistical Analysis

The statistical analysis and determination of the lethal concentrations 50 and 90 (LC_50_ and LC_90_), as well as the adjusted coefficient of determination (R^2^Adj), were performed using Origin Pro software (version 8.5) with a confidence interval set at 95% (*p* < 0.05).

## 3. Results and Discussion

### 3.1. Characterization of EOPb by GC–MS and NMR

In this study, the chemical composition of the volatile fraction of EO*Pb* was analyzed using the GC–MS technique (Table 1). A total of thirteen components were identified, accounting for 91.17% of the oil’s composition. The major component was citral comprising 64.58% of the oil, represented by two isomeric oxygenated monoterpenes: geranial (36.06%) and neral (28.52%). Other significant constituents included α-pinene (15.73%) and limonene (8.28%). This chemical profile is consistent with previous studies of the *Pectis* species collected from different regions of Brazil where citral content was reported to range from 81.6 to 86.5%, with α-pinene and limonene as secondary components [44].

To confirm the structure of the main compounds, EO*Pb* was analyzed by ^1^H, and ^13^C, and NMR spectroscopy (Table 2 and Table 3). The ^1^H NMR spectrum of EO*Pb* (Appendix A) displayed resonances indicative of a mixture of two isomers, The ^1^>H NMR spectrum of EO*Pb* (Appendix A) revealed resonances indicative of a mixture of two isomers, as evidenced by the variations in the aldehyde protons (H-9), which were identified by two doublets at δ 9.99 and 9.89. The integration of these signals yielded a neral-to-geranial ratio of 0.87:1.0. The proton signals resonating at δ 5.18 and 5.20 corresponded to the methine protons of cycloalkenes (H-3 and H-2), confirming the presence of α-pinene and limonene on EO*Pb*. In the ^13^C NMR spectra (Appendix A), the carbonyl signals (C-1) were observed at δ 191.2 and 191.5, respectively, to neral and geranial isomers. The signals at δ 144.5 (C-2) and 116.0 (C-3) were related to the *sp*^2^ carbons of α-pinene, while the signals at δ 133.6 (C-1), 120.6 (C-2), 163.8 (C-8), and 108.4 (C-9) corresponded to the *sp*^2^ carbons of limonene. The ^1^H and ^13^C NMR data aligned with the findings of Glamočlija et al. [54] and Farias et al. [55].

### 3.2. Development of the Empty Nanogels

To achieve the thermoresponsive behavior of the nanogels, certain combinations ranging from 5 to 20:0.1 to 0.3% (*w*/*w*) F127:974p were considered. Combinations of 5–10:0.1 to 0.3 resulted in liquid formulations at 5 and 30 °C, exhibiting no thermoresponsive behavior. Conversely, formulations comprising 20:0.1–0.3 remained liquid at 5 °C and transitioned to semi-solid at 30 °C, with a transition time of approximately 10 min from sol–gel. Notably, the combination of 20:0.1 yielded a low-viscosity formulation, whereas the combination of 20:0.3 resulted in high-viscosity formulations upon sensory evaluations. The optimized combination was determined to be 20:0.2 serving as the basis for incorporating the active ingredient EO*Pb* at various mass percentages, designated as nGF2002 with a rapid sol-to-gel transition Figure 3D.

### 3.3. Development of EOPb-Loaded Nanogels

Table 4 shows the combination of nGF2002 with different % (*w*/*w*) EO*Pb*. The preliminary stability evaluation of formulations GF1–GF11 provided valuable insights into their physical characteristics and potential suitability for nanogel development. Formulations GF1–GF4 presented as homogeneous, transparent solutions immediately after preparation, maintaining their integrity with no observable changes in physical appearance after 24 h of storage (Figure 3A). These formulations are promising candidates for further optimization and potential use in nanogel development, as their stability under refrigerated conditions indicates their suitability for practical applications. In contrast, formulations GF5–GF10 exhibited signs of instability, including slight turbidity upon preparation and phase separation after 24 h of storage, characterized by the appearance of an oily layer on the formulation’s surface (Figure 3B,C). The thickness of this oily layer increased from GF5 to GF9, indicating a decline in stability over time, and formulation adjustments to enhance stability and improve their potential for nanogel development. Formulation GF10 displayed poor stability, characterized by turbidity immediately post-preparation and triphasic separation after 24 h of storage. These observations suggest that GF10 may require significant modifications to address stability issues and render it suitable for further development. However, the GF11 formulation showed promising stability characteristics, maintaining an emulsion-like appearance after 24 h of storage (Figure 3C). This indicates its potential suitability for continued investigation and development of emulsion formulation. However, given that the study aimed to formulate nanogels (O/W), GF1-GF4 took precedence for subsequent accelerated stability investigations. This decision was made to streamline resources and focus on formulations demonstrating initial stability, potentially expediting the development process.

### 3.4. Physical Stability Assay of GF1–GF4

The results of the accelerated stability testing conducted on formulations GF1–GF4 provide valuable insights into the physical stability profiles of these semi-solid dosage forms. Adhering to the guidelines outlined by the US Pharmacopoeia (USP), our study scrutinized both the chemical integrity of the active pharmaceutical ingredients (API) and the physical characteristics of the formulations over a rigorous evaluation period [49]. Following USP standards, the chemical stability of compounded preparations is gauged by the maintenance of API concentration within 90–110% of the initial value recorded on day 0. Our findings reveal that formulations GF1–GF4 remained within this specified range throughout the 7-cycle evaluation period, indicating robust physical stability under the simulated storage conditions of alternating temperatures (5 and 32 °C). Furthermore, the absence of any visible alterations such as particle sedimentation or phase separation, as well as the preservation of organoleptic properties (odor and color), underscores the physical stability of the formulations. These observations were held irrespective of the storage conditions, signifying the formulations’ resilience to temperature fluctuations and light exposure over the 14-day accelerated stability testing period. The shelf-life assay was performed for 180 days under refrigerated conditions (5 °C) reaffirming the formulations’ sustained physical integrity and organoleptic characteristics. This prolonged stability underscores the suitability of the formulations for extended storage periods, particularly in environments where temperature control is a critical factor. The comprehensive evaluation of formulations GF1–GF4 demonstrates their robust physical and chemical stability profiles, positioning them as promising candidates for pharmaceutical applications. These findings contribute to the body of knowledge regarding the stability of semi-solid dosage forms and pave the way for their potential utilization in various therapeutic contexts. However, further studies incorporating additional parameters and longer-term evaluations are warranted to bolster our understanding and confidence in the stability and efficacy of these formulations. Considering the stable performance of nanogel formulations GF1–GF4 in stability assays, formulation GF4 was selected for further studies due to its higher content of EO*Pb*. This formulation has been designated as nGF2002*Pb* throughout this study (Figure 3D).

### 3.5. Characterization of nGF2002Pb

The CG analysis of nGF2002*Pb* was performed after the stability assay to investigate their chemical stability. A comparison of the chromatograms for EO*Pb* and nGF2002*Pb* reveals a high similarity of chromatograph profile, with the same elution order α-pinene, limonene, geranial, and neral. Interestingly, the chromatogram of nGF2002*Pb* shows a higher intensity of signals corresponding to the geranial and neral isomers related to the α-pinene and limonene compounds (Appendix A). Our interpretation is that being less hydrophobic, the isomers are accommodated in regions near the PEO chains of F127, making them more accessible. On the other hand, the greater interaction of α-pinene and limonene with the hydrophobic PPO chains allows for their preferential accommodation in the core of the F127 micelles, keeping them more protected from the external environment. The FTIR analysis of nGF2002 also supports these interpretations.

FTIR analysis is a highly effective tool often used to investigate the encapsulation process of active pharmaceutical ingredients (API) within polymeric nanoplatforms for drug delivery applications. By examining shifts or changes in band intensities and broadening of vibrations in the FTIR spectra, one can gain insights into properties such as miscibility and interactions between macromolecule-API chains. In the FTIR spectrum of EO*Pb* (Figure 4A), the medium-intensity absorption band between 2954–2870 cm^−1^ is associated with the symmetric and asymmetric stretching of C—H bonds. The sharp bands at 1442 and 1377 cm^−1^ are also associated with C—H bond vibrations, corresponding, respectively to the scissor and rocking vibrational modes The sharp, high-intensity band at 1674 cm^−1^ and the low-intensity band between 887–789 cm^−1^ are related, respectively, to the stretching and bending of C=C bonds in trisubstituted alkenes, supporting the presence of α-pinene and limonene in the chemical composition of EO*Pb*. The small shoulder observed at 1712 cm^−1^, characteristic of the carbonyl stretch of conjugated aldehydes, confirms the presence of the isomers neral and geranial in the composition of EO*Pb*. Figure 4B shows the FTIR spectrum for the nanogel nGF2002*Pb*.

Adding 1% of the EO*Pb* in the nGF2002 matrix significantly alters the absorption band values related to the F127/974p blend. The strong and sharp absorption band at 3676 cm^−1^ results from a pronounced red shift with Δν = 112 cm^−1^ from the band at 3564 cm^−1^ observed in the nGF2002 spectrum (Appendix A). This finding indicates that a decrease in energy is required for stretching O—H bonds, likely because EO*Pb* molecules can disrupt some of the F127/974p interactions to fit into the nGF2002*Pb* matrix. The band at 3384 cm^−1^ in nGF2002 is now shifted to 3340 cm^−1^ (44 cm^−1^ blue shift), likely due to the formation of new intermolecular hydrogen bonds between the blend F127/974p with the majority constituents of EO*Pb* leading to the interpretation that the more hydrophobic EO*Pb* molecules are preferentially accommodated in the PPO chains of F127. Part of this interpretation is based on the increased energy required for the anti-symmetric stretching of C—H bonds, as indicated by the increased intensity and blue shift (Δν = 19 cm^−1^) of the band from 2904 cm^−1^ (nGF2002) to 2885 cm^−1^ in the nGF2002*Pb* spectrum (Figure 4B), indicating a significant increase in the hydrophobic interactions of the system.

UV-Vis analyses performed on formulations GF1-GF4 at 5.0 × 10^−3^ g/mL in water support the encapsulation of EO*Pb* on the nGF2002 matrix (Appendix A). The absorption band observed at 240 nm corresponds to the presence of EO*Pb*. It is observed that, at the same concentration, an increase in the percentage of EO*Pb* in formulations GF1-GF4 results in a proportional increase in absorption at 240 nm, ensuring that EO*Pb* molecules remain solubilized and well-stabilized within the polymeric matrix of the nanogels.

The particle size and ζ potential of nGF2002*Pb* were quantitatively measured using the DLS technique. DLS has emerged as a pivotal tool for determining nanoparticle size distribution in a solution. By analyzing the fluctuations in scattered light intensity caused by Brownian motion, DLS provides valuable insights into the hydrodynamic diameter (D*_H_*) and polydispersity of nanogels assessing the homogeneity and stability of nanogel formulations. Complementary to DLS, ζ potential analysis offers a deeper understanding of nanogel behavior by evaluating their surface charge characteristics. The DLS analysis on the nGF2002*Pb* was conducted under two concentrations, maximum and minimum dilution in distilled water to ensure that equipment could accurately measure the particle size and ζ potential properties in solution. The results of the DLS measurements revealed notable differences. Under maximum dilution, the D*_H_* of the particles showed an average value of 30.44 nm and PDI of 0.54, indicating a greater dispersion of particles in the solution (Figure 4C). Typically, nanoparticles intended for drug delivery fall within the size range of 10 to 200 nm. Therefore, selecting the appropriate nanoparticle size is crucial for optimizing the effectiveness of drug delivery systems. It is well-documented that F127 has a higher propensity to self-organize in dilute solutions, typically forming spherical aggregates. This range is chosen because nanoparticles within this size range exhibit favorable characteristics such as enhanced cellular uptake, prolonged circulation time in the bloodstream, and improved biodistribution.

D*_H_* significantly increased to 429.3 nm, and PDI of 0.27 under minimum dilution (Appendix A). At higher concentrations, F127 can self-assemble into a morphological complex. Larger aggregates may be interconnected by branched architecture, named worm-like micelles, leading to stable aggregates with even larger sizes, as evidenced in previous studies [56]. Additionally, at higher concentrations, the nanogels can remain intact for extended periods in applications requiring controlled release of active pharmaceutical ingredients. This behavior is attributed to the robust structure formed by worm-like micelles, which facilitates sustained and controlled release of the active substances. Studies show that due to their branched architecture and ability to organize in three-dimensional network forms, nanogels can maintain the stability of the pharmaceutical for a long time, releasing the active ingredient gradually and over a prolonged period. This characteristic is particularly beneficial for treatments requiring extended release, enhancing therapeutic efficacy, and reducing the need for frequent administration. The measurements of ζ potential did not show significant changes in values across these concentrations. Interestingly, both dilution conditions exhibited ζ potential values close to zero (Figure 4D). Despite the proximity of the ζ potential to zero, indicating a lack of strong electrostatic repulsion between particles, the system’s stability was not noticeably affected.

### 3.6. In Vitro Assays Against Aedes aegypti Larvae

The results presented in Table 5 underscore the potent larvicidal efficacy of the nGF2002*Pb* nanogel against *Aedes aegypti* larvae at varying concentrations in EO*Pb*-loaded nGF2002, ranging from 5 to 500 μg/mL, across 24 h and 48 h treatment intervals. The mortality rates, expressed as percentages with accompanying standard deviations, were evaluated for a sample size of 10 larvae per concentration, providing robust insights into the concentration-dependent effects of the nanogel formulation. The larvicidal activity observed at the lowest concentrations was minimal, with mortality rates of 1.0% ± 3.2 after 24 h and 1.0–2.0% ± 3.2–4.2 after 48 h (Entries 1 and 2). These findings suggest that the nanogel at such low dosages is insufficient to disrupt critical biological functions in the larvae, highlighting the importance of optimizing the concentration for effective vector control. The application of nGF2002*Pb* at 100 μg/mL resulted in a moderate increase in mortality, reaching 11.0% ± 12.9 at 24 h and 16.0% ± 13.5 at 48 h (Entry 3). Although a temporal increase in efficacy was noted, the overall larvicidal activity remained suboptimal. This observation underscores the need for a threshold concentration that can trigger significant physiological disturbances in the larvae, leading to mortality. A marked improvement in larvicidal efficacy was achieved at higher concentrations, with the 250 μg/mL of nGF2002*Pb* causing 67.0% ± 9.5 mortality at 24 h and 89.0% ± 3.2 at 48 h (Entry 4). The efficacy was even more pronounced at 500 μg/mL, with mortality rates of 96.0% ± 7.0 after 24 h and 100.0% ± 0.0 after 48 h (Entry 5). These results indicate that the nGF2002*Pb* demonstrates a pronounced concentration-dependent larvicidal effect. This is likely attributed to the enhanced bioavailability and sustained release of EO*Pb*, which disrupts the physiological processes of the larvae. The absence of larval mortality in the negative controls, which included water and nGF2002 without EO*Pb*, confirmed that the observed mortality in the experimental groups is attributable solely to the bioactive effects of the EO*Pb* (Entry 6). This highlights the specificity of the nanogel formulation and its potential as an effective larvicidal agent. The positive control group treated with temefos (Entry 7), a widely recognized larvicide, achieved 100% mortality at both time points, validating the experimental conditions and providing a benchmark for assessing the efficacy of the nGF2002*Pb* nanogel. The comparable performance of the highest concentration of nGF2002*Pb* with that of temefos underscores the promising potential of nanogel as an alternative larvicidal intervention. The findings from this study suggest that the nGF2002*Pb* nanogel represents a viable and eco-friendly alternative for controlling *Aedes aegypti* larvae, particularly at higher concentrations. The concentration-dependent efficacy observed indicates that formulation optimization could enhance its practical applicability in vector control programs.

Figure 5 shows the results of lethal doses (LC_50_ and LC_90_) after 24 and 48 h of treatment with nGF2002*Pb*. The results indicate that the nGF2002*Pb* nanogel is highly effective in controlling *Aedes aegypti* larvae, with a significant reduction in lethal concentrations over the exposure time. After 24 h, LC_50_ was 199.5 μg/mL, and LC_90_ was 392.0 μg/mL, while after 48 h, these values decreased to 184.5 μg/mL and 253.4 μg/mL, respectively. The decrease in lethal concentrations over time suggests increasing larval sensitivity to the nanogel, possibly due to higher absorption or a cumulative effect of the treatment. The R^2^ values, very close to 1, indicate high precision in the dose–response models, reinforcing the reliability of the obtained data. These results highlight the potential of the nGF2002*Pb* nanogel as a promising tool in the fight against the mosquito vector of diseases. Future studies should focus on the mechanical aspects of larval mortality induced by the nanogel and field trials to validate its effectiveness in real-world settings. This is crucial to better understand the cytotoxic processes of the nanogel against *Aedes aegypti* larvae, with results to be published in forthcoming research. Integrating these nanotechnological approaches has the potential to significantly enhance mosquito control strategies, making them more sustainable and targeted, and effectively mitigating the spread of arboviral diseases. The larvicidal efficacy of nGF2002*Pb* corroborates findings in the literature that emphasize characteristics such as the controlled release of volatile constituents from essential oils, as well as their structural stability. In this context, the larvicidal performance of hydrogel spheres composed of sodium alginate (SA) combined with a Pickering emulsion (PE) of white thyme EO was evaluated against *Aedes albopictus* larvae, a vector for significant arboviruses in rural areas. A controlled release of the EO was observed, directly correlating with an increase in larvicidal activity over 48 h [57]. Likewise, the stability of the *Eucalyptus globulus* EO nanogel conferred high larvicidal efficiency against *Anopheles stephensi* larvae (LC_50_ 32 μg/mL), a prominent domestic species of malaria vector mosquito [58].

Although a direct comparison of the LC_50_ results published in the literature and the values obtained in this study for GF2002*Pb* may show discrepancies, it is essential to consider that the systems involved are considerably distinct. First, regarding the excipients of the formulations, many studies describing the development of nanogels use non-ionic surfactants, such as polysorbates for the solubilization of EOs. It is important to emphasize that this class of surfactants has significantly high critical micelle concentrations (CMCs), for example, Tween-80 has a CMC of up to 2.2 mM at 298.15 K, a value much higher than that of Pluronic F127, which is 0.34 mM at the same temperature [59]. This may influence the precipitation of the formulated content when administered in biological environments, which, in turn, could explain the higher LC_50_ values reported in these studies [42,43,57,58]. Additionally, the application of GF2002 (empty nanogel) did not show any cytotoxic effect on *Aedes aegypti* larvae, reinforcing the non-toxic profile of F127, approved by the FDA as a safe excipient. It is also relevant to consider the differences in the chemical profile of each investigated EO, as variations in their composition may directly reflect in the observed LC_50_ values, justifying the discrepancies compared to the literature data. Finally, this work represents an advancement in the state of the art by employing, for the first time, an eco-friendly thermoresponsive nanogel based on the F127 copolymer and Carbopol 974p for larvicidal purposes.

## 4. Conclusions

The findings from this study suggest that the nGF2002*Pb* nanogel represents a viable and eco-friendly alternative for the control of *Aedes aegypti* larvae, particularly at higher concentrations. The characterization methods CG, FTIR, and UV-Vis confirmed the stable incorporation of EO*Pb* within the nanogel matrix. The results of the DLS measurements revealed a hydrodynamic diameter (D*_H_*) of 30.44 nm and a polydispersity index (PDI) of 0.54, indicating a greater dispersion of particles in the solution. Specifically, the in vitro assays demonstrated a mortality rate ranging from 96.0 to 100.0% between 24 and 48 h at a concentration of 500 μg/mL. This concentration-dependent efficacy suggests that further optimization of the formulation could enhance its practical applicability in vector control programs. In light of these results, the next steps of this study will involve assessing the cytotoxic profile of the nG2002*Pb* nanogel using zebrafish (*Danio rerio*) as a model organism, in collaboration with the Applied Toxicology Laboratory (LETA) at the Butantan Institute in São Paulo, SP. This approach will ensure the safety of the material concerning non-target aquatic organisms. Additionally, future studies should focus on the mechanical aspects of larval mortality induced by the nanogel, as understanding the mechanisms of action will provide valuable insights into the effectiveness of nGF2002*Pb*. Furthermore, conducting field trials is essential to validate the nanogel’s effectiveness in real-world settings, ensuring that laboratory results translate into practical applications. The integration of such nanotechnological approaches could significantly enhance sustainable and targeted mosquito control strategies, ultimately mitigating the spread of arboviral diseases and addressing the pressing public health challenges posed by *Aedes aegypti* and the diseases it transmits.

## Figures and Tables

**Figure 1 pharmaceutics-16-01337-f001:**
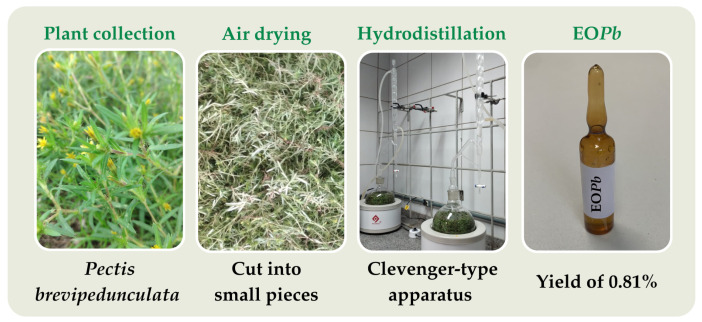
Experimental sequence for the extraction procedure of EO*Pb* involved air drying of the plant material, trituration, and hydrodistillation under controlled conditions. The essential oil yield was 0.81%, and the collected oil was stored appropriately for subsequent analysis.

**Figure 2 pharmaceutics-16-01337-f002:**
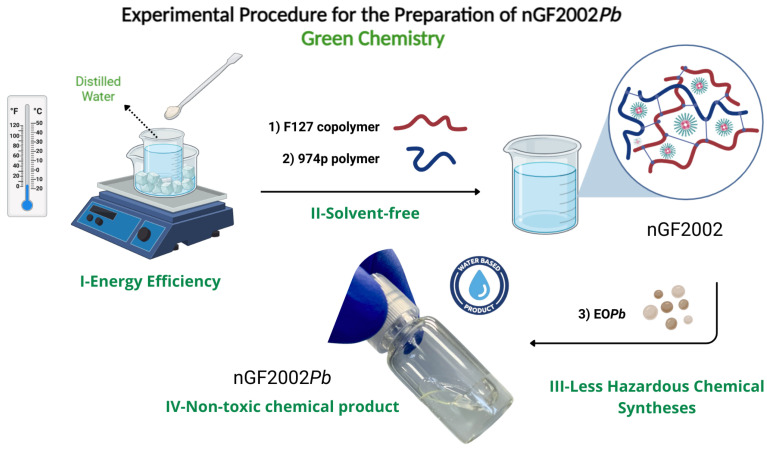
Schematic illustration of the experimental sequence for the preparation of the nGF2002*Pb*. Green chemistry procedures are presented in the experimental steps.

**Figure 3 pharmaceutics-16-01337-f003:**
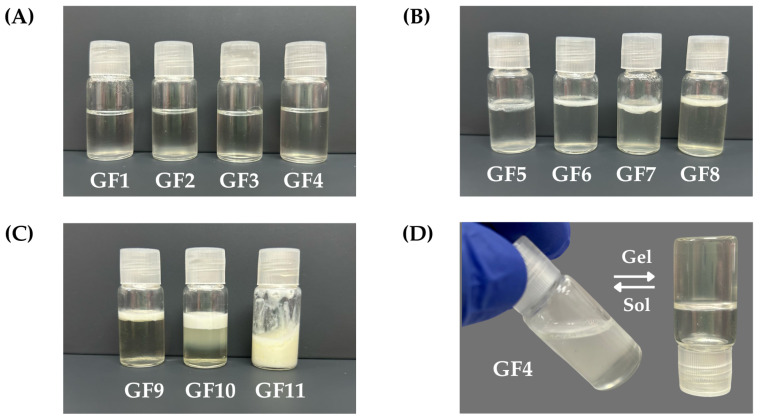
Photos of nanogel formulations prepared in different compositions % (*w*/*w*) of F127, 974p, H_2_O, and EO*Pb* according to Table 4. (**A**) GF1-GF4 show stable formulations; (**B**) GF5 to GF8 exhibit phase separation; (**C**) GF9 and GF10 show phase separation, and GF11 exhibits stability in the form of a nanoemulsion. (**D**) Thermoresponsive behavior was shown for the GF4 formulation.

**Figure 4 pharmaceutics-16-01337-f004:**
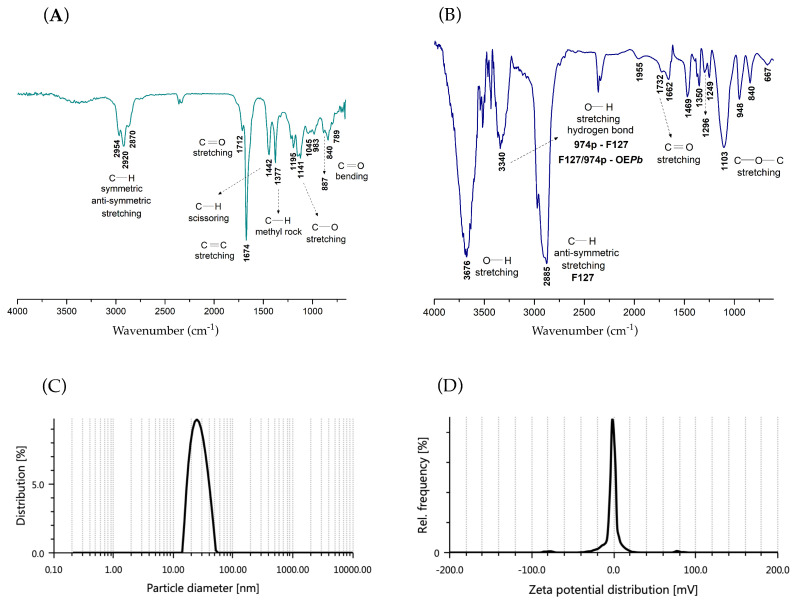
FTIR spectra: (**A**) EO*Pb* and (**B**) nGF2002*Pb* and DLS analysis of the nGF2002*Pb*: (**C**) Particle size distribution under high dilution conditions, showing a D*_H_* of 30.18 nm and PDI of 0.54, and (**D**) ζ potential of −1.6 mV.

**Figure 5 pharmaceutics-16-01337-f005:**
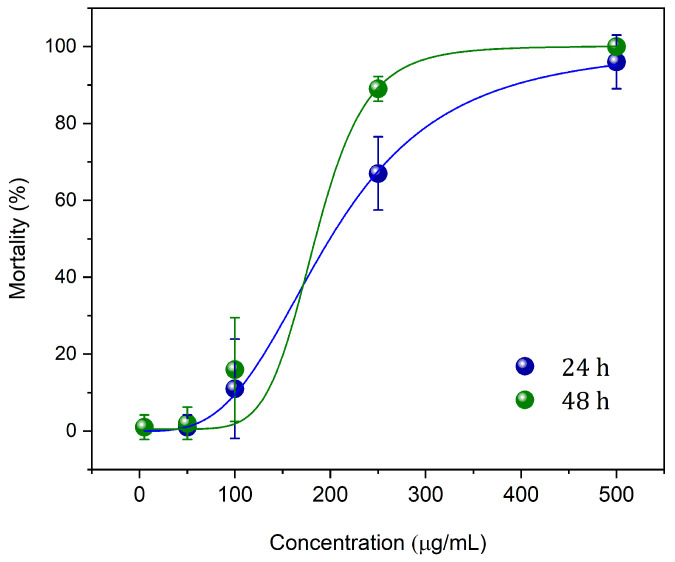
Lethal concentrations LC_50_ and LC_90_ for *Aedes aegypti* larvae. After 24 h of treatment with nGF2002*Pb* nanogel, the LC_50_ was 199.5 ± 5.6 μg/mL and the LC_90_ was 392.0 ± 16.6 μg/mL, with an R^2^ of 0.9992. After 48 h of treatment, the LC_50_ decreased to 184.5 ± 8.2 μg/mL and the LC_90_ to 253.4 ± 7.6 μg/mL, with an R^2^ of 0.9989.

**Table 1 pharmaceutics-16-01337-t001:** GC–MS analysis of OE*Pb*: the retention index (RI), molecular formula (MF), and relative percentage (%).

Number	Compound	RI	RI ^a^	MF	%
Monoterpene hydrocarbons (24.5%)					
1	**α-Pinene**	**939**	**932**	**C** _10_ **H** _16_	**15.73**
2	Camphene	949	946	C_10_H_16_	0.1
3	β-Pinene	975	974	C_10_H_16_	0.28
4	**Limonene**	**1025**	**1024**	**C** _10_ **H** _16_	**8.28**
5	(*E*)-β-Ocimene	1042	1044	C_10_H_16_	0.11
Oxygenated monoterpenes (66.42%)					
6	Linalool	1089	1095	C_10_H_18_O	0.37
7	6-Camphenol	1108	1111	C_10_H_18_O	0.1
8	*E*-isocitral	1165	1160	C_10_H_18_O	1.02
**9**	**Neral**	**1230**	**1235**	**C**_10_H_18_O	**28.52**
10	Piperitone	1240	1249	C_10_H_18_O	0.17
**11**	**Geranial**	**1262**	**1264**	**C**_10_H_18_O	**36.06**
Fatty acid derivative					
12	Geranyl acetate	1373	1379	C_12_H_20_O_2_	0.18
Sesquiterpene hydrocarbon					
13	β-Elemene	1407	1398	C_12_H_24_	0.25

^a^: Reference [44].

**Table 2 pharmaceutics-16-01337-t002:** NMR data for ^1^H and ^13^C (500 and 125 MHz, respectively; CDCl_3_) of neral and geranial present in EO*Pb*; a comparison with literature data. (δ) chemical shifts.

	EO*Pb*				Literature ^a^			
	**Neral**		**Geranial**		**Neral**		**Geranial**	
**Position**	$\delta_{H}$	$\delta_{C}$	$\delta_{H}$	$\delta_{C}$	$\delta_{H}$	$\delta_{C}$	$\delta_{H}$	$\delta_{C}$
1	9.99	191.2	9.89	190.7	9.99	191.5	0.90	109.6
2	5.88	127.4	5.88	128.6	5.88	127.1	5.88	128.7
3	-	163.8	-	163.8	-	163.9	-	163.9
4	2.20	40.6	2.59	32.6	2.20	40.3	2.59	32.3
5	2.17	25.7	2.17	27.9	2.17	25.5	2.17	27.8
6	5.18	122.6	5.09	122.3	5.16	122.4	5.10	122.1
7	-	132.9	-	133.6	-	132.9	-	133.4
8	1.68	25.6	1.68	25.6	1.68	25.3	1.68	25.3
9	1.61	17.7	1.61	17.5	1.61	17.4	1.61	17.4
10	1.59	17.5	1.98	25.0	1.99	17.2	1.99	24.7

^a^: Reference: [54].

**Table 3 pharmaceutics-16-01337-t003:** NMR data for ^1^H and ^13^C (500 and 125 MHz, respectively; CDCl_3_) of α-pinene and limonene present in EO*Pb*, and a comparison with literature data’ (δ) chemical shifts.

	EO*Pb*				Literature ^a^			
	α **-Pinene**		**Limonene**		α **-Pinene**		**Limonene**	
**Position**	$\delta_{H}$	$\delta_{C}$	$\delta_{H}$	$\delta_{C}$	$\delta_{H}$	$\delta_{C}$	$\delta_{H}$	$\delta_{C}$
1	1.98	47.0	-	133.6	1.93	47.0	-	133.6
2	-	144.5	5.40	120.6	-	145.0	5.45	120.6
3	5.18	116.0	(2.35; 2.31)	30.8	5.20	116.0	(2.28; 2.08)	30.8
4	(2.23; 2.21)	31.2	1.65	41.1	(2.23; 2.21)	31.2	1.67	41.0
5	2.04	40.7	(1.61; 1.59)	30.6	2.06	40.7	(1.67; 1.49)	30.5
6	-	37.9	2.08	27.9	-	37.9	2.08	27.9
7	(1.61; 1.59)	31.4	1.73	23.4	(1.61; 1.55)	31.4	1.79	23.4
8	1.26	26.3	-	163.8	1.28	26.3	-	159.2
9	0.83	20.8	4.70	108.4	0.83	20.8	4.77	108.3
10	1.66	22.9	1.65	22.5	1.67	22.9	1.71	20.6

^a^: Reference [55].

**Table 4 pharmaceutics-16-01337-t004:** Different compositions % (*w*/*w*) of EO*Pb* and water, were combined to F127:974p (20:02%, *w*/*w*) to achieve GF1-GF11.

	Components				
Code	Water	F127	974p	EO*Pb*	Stability
GF1	79.7	20	0.2	0.125	S ^a^
GF2	79.6	20	0.2	0.25	S ^a^
GF3	79.3	20	0.2	0.5	S ^a^
GF4	78.8	20	0.2	1	S ^a^
GF5	78.6	20	0.2	1.25	PS
GF6	78.3	20	0.2	1.5	PS
GF7	77.8	20	0.2	2	PS
GF8	77.3	20	0.2	2.5	PS
GF9	74.8	20	0.2	5	PS
GF10	69.8	20	0.2	10	PS
GF11	59.8	20	0.2	20	S ^b^

S: Stable; PS: Phase Separation. ^a^ Nanogel; ^b^ Nanoemulsion.

**Table 5 pharmaceutics-16-01337-t005:** Larvicidal efficacy of nGF2002*Pb* against *Aedes aegypti* larvae at 24 and 48 h post-treatment represented in % mortality ± Standard Deviation (SD) for a n = 10.

Entry	Treatment μg/mL	Mortality % ± SD	
	nGF2002*Pb* ^a^	24 h	48 h
**1**	5	1.0 ± 3.2	1.0 ± 3.2
**2**	50	1.0 ± 3.2	2.0 ± 4.2
**3**	100	11.0 ± 12.9	16.0 ± 13.5
**4**	250	67.0 ± 9.5	89.0 ± 3.2
**5**	500	96.0 ± 7.0	100.0 ± 0.0
**6**	**Water**	0.0 ± 0.0	0.0 ± 0.0
**7**	**nGF2002 ^b^**	0.0 ± 0.0	0.0 ± 0.0
**8**	**Temefos ^c^**	100.0 ± 0.0	100.0 ± 0.0

^a^ Concentration in EO*Pb*; ^b^ Negative control tested at 1 × 10^5^ μg/mL. ^c^ Positive control tested at 100 μg/mL

## Data Availability

The original contributions presented in the study are included in the article/Appendix A, further inquiries can be directed to the corresponding author.

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
