# Peer review of "Development of an Eco-Friendly Nanogel Incorporating Pectis brevipedunculata Essential Oil as a Larvicidal Agent Against Aedes aegypti"

_pharmaceutics, 2024, doi:10.3390/pharmaceutics16101337_

Round 1

Reviewer 1 Report

Comments and Suggestions for Authors

Thank you for submitting the article titled: Development of an Eco-Friendly Nanogel Incorporating Pectis brevipedunculata Essential Oil as Larvicidal Agent Against Aedes aegypti. The article presents an interesting topic and is well-designed; however, I am sending a few comments below that will improve the quality of the article and its reception by readers:

  1. Please separate the grouped references into individual citations rather than listing them together. In some places in the introduction, five references are assigned to two sentences – while this can be appropriate when citing specific examples, in such a general statement, it comes across as artificially inflating the number of references.
  2. The introduction lacks a paragraph at the end describing what was done in the study and what analyses were performed.
  3. In the materials section, please add a sentence indicating whether the purchased reagents were modified in any way.
  4. Add a diagram to subsection 2.3.
  5. The synthesis should be described before the methodology, and in this section, there should be a table with the compositions of the obtained formulations.
  6. The charts for the results should be presented in the subsection where they are discussed, not in a different one – please adjust this.
  7. Please expand the conclusions, including the numerical values obtained from the discussion.

Author Response

We would like to express our sincere gratitude to the reviewers for their insightful comments regarding our work, Pharmaceutics-3249322. It is a pleasure to have the expertise of the reviewers, who will be instrumental in enhancing the quality of the final version of the manuscript. Your valuable feedback is greatly appreciated and will significantly contribute to the improvement of our research and its reception by readers. Please note that the corrections have been highlighted in yellow in the text.

Comments 1: Please separate the grouped references into individual citations rather than listing them together. In some places in the introduction, five references are assigned to two sentences – while this can be appropriate when citing specific examples, in such a general statement, it comes across as artificially inflating the number of references.

Response 1: As per the reviewer's suggestion, the references have been separated and distributed throughout the different topics, now organized into distinct sections across various paragraphs. This redistribution ensures a more precise and relevant placement of citations, avoiding the impression of artificially inflating the number of references, as also noted by the reviewer.

Comments 2: The introduction lacks a paragraph at the end describing what was done in the study and what analyses were performed.

Response 2: In response to the reviewer's observation, a new paragraph has been added at the end of the introduction to describe the work conducted in the study and the analyses performed, as suggested.

Comments 3: In the materials section, please add a sentence indicating whether the purchased reagents were modified in any way.

Response 3: According to the editor's suggestion, this information has now been included in the revised version of the manuscript

Comments 4: Add a diagram to subsection 2.3.

Response 4: A diagram illustrating the sequential steps of the extraction process of EOPb has been created and labeled as Figure 1.

Comments 5: The synthesis should be described before the methodology, and in this section, there should be a table with the compositions of the obtained formulations.

Response 5: The methodology for preparing the nanogels is described in the methodology section (2.8). Since the objective was to outline the preliminary stabilities of formulations GF1 to GF11, we found it more convenient to present this in the form of results and discussions. However, if the reviewer believes that this should still be altered, we can consider a better way to represent it.

Comments 6: The charts for the results should be presented in the subsection where they are discussed, not in a different one – please adjust this.

Response 6: The results have been appropriately inserted into the respective section, as suggested.

Comments 7: Please expand the conclusions, including the numerical values obtained from the discussion.

Response 7: The conclusion has been expanded, and the numerical results have been included as suggested.

Reviewer 2 Report

Comments and Suggestions for Authors

Review of manuscript pharmaceutics-3249322

In this work, the authors have prepared nanogel formulations of Pectis brevipedunculata essential oil, evaluated the stabilities, and determined the toxicity to larvae of Aedes aegypti. As a preliminary study, this work provides evidence for the potential utility in essential oil-containing nanogels for mosquito control. Publication is recommended after some minor points to consider.

Abstract: The abstract provides a useful and concise summary of the work. There is only one question: Line 10, 500 g/mL is incredibly high. Presumably, the authors mean 500 μg/mL.

Introduction: The Introduction is a long run-on paragraph. It could be split into several paragraphs based on topics. For example, start new paragraphs on line 30 (“Among the most…”), line 36 (“Plant-based compounds…”), line 46 (“The integration…”), line 57 (“Nanogels and nanoemulsions…”), and line 72 (“The Pectis genus…”). Please also include a brief introduction of Pectis brevipedunculata, including complete name and family, in this last paragraph.

Materials and Methods: The Materials and Methods section is complete and detailed enough to allow other researchers to repeat the experiments. There are only some minor points to consider. Line 89, Aedes aegypti should be in italics. Lines 111-112 and in several places throughout the manuscript, Place a space between a measurement and its units.

Results: Since there is no separate Discussion section, should this be “Results and Discussion”? Line 202 and thereafter: How can you know it is the D-enantiomer of limonene? It would be better to just use limonene. Table 1: Limonene rather than D-Limonene, Geranyl acetate [two words], β-Elemene [English]; footnote, be sure to use the correct formatting for references. Tables 2 and 3: Use the correct formatting for references. Line 294: “…between 2954 and 2870 cm-1…” Line 298: “…between 887 and 789 cm-1…” Line 364: “…across 24-h and 48-h treatment intervals” [insert hyphens]. Lines 403-404: These LC50 values are not particularly compelling; the potency of the essential oil-containing nanogels should be discussed in comparison with previous reports on essential oils and nano formulations. Line 412: Aedes aegypti should be in italics.

Conclusions: Future studies should also address toxicity to these nanogels toward non-target aquatic organisms.

References: The page numbers or manuscript numbers are missing for references 1, 6, 19, 20, 21, 23, 29, 30, 32, 33, 41, and 52. Are there URLs available for references 49 and 50? Scientific binomials in the references should be in italics.

Author Response

We would like to express our sincere gratitude to the reviewers for their insightful comments regarding our work, Pharmaceutics-3249322. It is a pleasure to have the expertise of the reviewers, who will be instrumental in enhancing the quality of the final version of the manuscript. Your valuable feedback is greatly appreciated and will significantly contribute to the improvement of our research and its reception by readers. Please note that the corrections have been highlighted in yellow in the text.

Reviewer 2: In this work, the authors have prepared nanogel formulations of Pectis brevipedunculata essential oil, evaluated the stabilities, and determined the toxicity to larvae of Aedes aegypti. As a preliminary study, this work provides evidence for the potential utility of essential oil-containing nanogels for mosquito control. Publication is recommended after some minor points to consider. 

Comments 1: Abstract: The abstract provides a useful and concise summary of the work. There is only one question: Line 10, 500 g/mL is incredibly high. Presumably, the authors mean 500 μg/mL.

Response 1: The reviewer’s observation is correct, and the correction has been made.

Comments 2: Introduction: The Introduction is a long run-on paragraph. It could be split into several paragraphs based on topics. For example, start new paragraphs on line 30 (“Among the most…”), line 36 (“Plant-based compounds…”), line 46 (“The integration…”), line 57 (“Nanogels and nanoemulsions…”), and line 72 (“The Pectis genus…”). Please also include a brief introduction of Pectis brevipedunculata, including complete name and family, in this last paragraph.

Response 2: As per the reviewer's suggestion, the introduction has been separated and distributed throughout the different topics, now organized into distinct sections across various paragraphs. Furthermore, a brief introduction of Pectis brevipedunculata was included as suggested.  

Comments 3: Materials and Methods: The Materials and Methods section is complete and detailed enough to allow other researchers to repeat the experiments. There are only some minor points to consider. Line 89, Aedes aegypti should be in italics. Lines 111-112 and in several places throughout the manuscript, Place a space between a measurement and its units.

Response 3: The corrections were made throughout the text according to the suggestions.

Comments 4: Results: Since there is no separate Discussion section, should this be “Results and Discussion”? Line 202 and thereafter: How can you know it is the D-enantiomer of limonene? It would be better to just use limonene. Table 1: Limonene rather than D-Limonene, Geranyl acetate [two words], β-Elemene [English]; footnote, be sure to use the correct formatting for references. Tables 2 and 3: Use the correct formatting for references. Line 294: “…between 2954 and 2870 cm-1…” Line 298: “…between 887 and 789 cm-1…” Line 364: “…across 24-h and 48-h treatment intervals” [insert hyphens]. Lines 403-404: These LC50 values are not particularly compelling; the potency of the essential oil-containing nanogels should be discussed in comparison with previous reports on essential oils and nano formulations. Line 412: Aedes aegypti should be in italics.

Response 4: The reviewer’s comments have been duly acknowledged. The section has been renamed to "Results and Discussions," and the nomenclature "D-limonene" has been replaced with "limonene." Additionally, all other corrections have been implemented as requested. Furthermore, the discussion now includes a comparative analysis of the LC50 values of previous reports on essential oils and nanofabricated formulations, as suggested.

Comments 5: Conclusions: Future studies should also address toxicity to these nanogels toward non-target aquatic organisms.

Response 5: The conclusion has been revised by the suggestions provided by the reviewer.

Comments 6: References: The page numbers or manuscript numbers are missing for references 1, 6, 19, 20, 21, 23, 29, 30, 32, 33, 41, and 52. Are there URLs available for references 49 and 50? Scientific binomials in the references should be in italics.

Response 6: All references have been revised in the corrected version of the manuscript.